# Design of a Smart Barrier to Internal Flooding

Jorge Muñoz-Caballero, Diego Vergara *, Pablo Fernández-Arias and Álvaro Antón-Sancho

Technology, Instruction and Design in Engineering and Education Research Group, Catholic University of Ávila, C/Canteros s/n, 05005 Ávila, Spain

* Correspondence: diego.vergara@ucavila.es

**Abstract:** Increasingly, and with greater frequency, humanity is experiencing violent weather storms, which cause innumerable human and economic losses. Among the most frequent climatic storms that cause considerable losses are floods. Usually, domestic anti-flood systems are not autonomous; they require human intervention. This article presents Smart Flood Barrier (SFB), which is a fully automated system using sensors and composed of hinged lintels that can be installed on any door of any commercial brand. SFB is responsible for diverting the flow of fluid towards the center of a street with a certain slope, generating a "dry zone" near the entrance door to the house. This device also has a barrier installed in front of the door, which will be responsible for hermetically sealing the lintels, preventing the passage into the interior of the house if it is possible for the fluid to flow in the opposite direction to the direction of fluid flow.

**Keywords:** natural disaster; flood; emergency response; barrier; single-family homes; automation





## 1. Introduction

On countless occasions, and with increasing frequency, society is experiencing extraordinary phenomena that cause great material and human damage [1]. Extraordinary risks to people and property are divided between those caused by natural forces (earthquakes, floods, hurricanes, storms, etc.), and those caused by terrorism, rebellion, riots, riots, etc. In developing African countries, such as Ghana, there have been an estimated a total of 517 fatalities, USD 615,192,000 estimated damages, and 5,016,292 lives affected from the year 1900 to the first half of 2021 [2]. The devastating effects of Hurricane Katrina in New Orleans in 2005 and Hurricane Sandy in the New York metropolitan area in 2012 exemplified the extent to which today's cities are vulnerable and susceptible to losses due to natural catastrophes [3]. Within these meteorological phenomena and the damage produced against property and real estate are floods [4–7], which are feared for their significant consequences, including flooding inside homes and the enormous economic losses that this entails [8,9].

The consequences of climate change have global, far-reaching effects on the social and economic life of countries [10,11]. Weather systems are changing, sea levels are rising, and weather events are becoming more extreme. Their effects are disrupting national economies and affecting lives [12]. In the specific case of floods, their effects are so damaging that the fight against them has been included within the Sustainable Development Goals (SDGs), namely in SDG 13 on climate action [13]. According to the Insurance Compensation Consortium and the Geological and Mining Institute of Spain, floods are the natural catastrophe that generates the greatest damage in the country, with an estimated cost being around EUR 800 million per year [14]. In the period 1971–2016, according to Consortium statistics, 44.6% of the files processed were due to flood damage, which accounted for 62.0% of the total compensation, which, on average, amounts to about USD 130 million each year. However, the consequences of floods are even more critical in rural areas and underdeveloped countries [15]. The high vulnerability of these regions and the need to fulfil

the SDGs make it necessary to develop systems that can reduce the drastic consequences of floods.

Flood hazards cause severe damage to buildings and infrastructure around the world with an annual loss bill in terms of billions of dollars. The use of temporary internal flood barriers is a relatively quick mitigation measure for the consequences of internal flooding and gives households time to perform rapid mitigation actions such as elevating water-sensitive components, moving vehicles, or constructing other temporary barrier system [16]. These times to respond to the consequences of internal flooding are even longer if fixed barrier systems are in place. Likewise, these types of systems reduce the consequences of the flood hazards.

In the case of internal flooding, there are currently patents marketed by different distributors that are related to the prevention of damage caused by natural flooding. Examples include: (i) those of Anti-Flood Barriers [17], which market modular anti-flood barriers from 40 cm to 100 cm through quick and easy manual assembly; (ii) Floodark [18], with modular barriers without the need for tools for manual assembly; (iii) the barriers of Lakeside Flood Solutions [19], which markets automatic anti-flood gates using a flood chamber and float system that are responsible for causing the gate to rise vertically when the level in the street rises; (iv) those of LCF Technologies [20], with dimensions that embrace all types of doors and windows, have classic types of gates (with manual assembly and disassembly)—integral gates for use as a door, modular type gates, and gates for businesses integrated into the closures; and finally, (v) those of Flood Control International [21], which has removable temporary barriers, glass barriers, self-raising gates by hydraulic power, deployable barriers, containment barriers for chemical spills, among others. As shown in Table 1 below, in view of the advantages that the SFB innovation offers over previous patents, the main objective of the prototype developed in this work, called Smart Flood Barrier (SFB), is to minimize or remove the damage caused by floods, flash floods, etc., inside a house or premises located on a street with a certain slope, all in a fully automatic way.

**Table 1.** Significant advantages SFB compared to previous patents.

| Patent | SFB Comparative Advantage |
|---|---|
| Anti-Flood Barriers [17] | The flood forecast must be known in order to pre-assemble the barrier. However, SFB is a stand-alone system and can be controlled remotely. |
| Floodark [18] | This is a manually assembled barrier, requiring installation prior to flooding. It is based on a modular system in which a certain number of modules can be selected to cover the desired height. The joints present watertightness problems. SFB is a complete barrier, without intermediate joints, and can be manufactured to the height needed by the client. |
| Lakeside Flood Solutions [19] | Similar to the SFB system, except that in this case, the barrier is raised mechanically by flooding the moat, with the barrier having a flotation system at the bottom. This system has the disadvantage of not filling the moat properly if the inlet nozzle is blocked by mud or objects dragged by the flood. |
| LCF Technologies [20] | Containment system by manual installation, with an automatic system for shop doors that are installed on the closing shutter. This system is not viable in this case as many shops do not require locking systems and their entrances and shop windows are visible to the public. In addition, they have a rubber-butyl system that is similar to SFB to hermetically seals the perimeter, but which can be degraded by friction as the barrier moves. In SFB, the chamber rises deflated and inflates once the barrier is fully deployed. |
| Flood Control International [21] | This system has numerous types and models for different applications. In the case of its automatic barrier, it is presented as a folding barrier, installed on the ground, of great height and weight, mainly indicated for garage doors, preventing them from being flooded by the water of a flood. Its lifting system is formed by a hydraulic cylinder and chains that, when activated by sensors or remotely, close the door preventing the entry of water. As SFB is smaller in size, its application is focused on entrance doors of single-family dwellings, or doors of communal dwellings. Furthermore, with regard to electricity consumption, this is lower due to the lower weight of our system. Finally, SFB is more environmentally friendly as leakage of hydraulic oil can cause accidents and environmental pollution. |

To achieve this goal, SFB offers several advantages over previous patents: (i) it can be installed both in entrance doors to single-family dwellings and in newly built community dwellings, and even in existing doors by means of a small refurbishment work; (ii) it does not need previous assembly before flooding by the user, as it happens with most current systems, because it is a fully automatic system and, in the case of a possible flood, it acts intelligently by diverting and preventing the entry of water into the property, thus sustaining protection; and (iii) it does not need the presence of the user for its installation. In addition, the system has manual control so that the user can deploy and collect the lintels and vertical door barrier if necessary.

After this analysis, SFB aims to protect people's assets, not only the material ones, which are relatively easy to replace, but also those assets that have an emotional or sentimental component for the affected party; these are usually precisely those that represent the greatest loss for those affected by these events. Damage reduction covers both human and material assets, as well as damage to buildings themselves. Finally, SFB aims to reduce the costs incurred by insurers in compensating those affected by this type of weather phenomenon.

## 2. Materials and Methods

The objective of the present invention is to protect the material goods located inside buildings and dwellings, as well as the people who may be inside them. For this purpose, it is necessary to carry out an initial design considering the standard measures of the doors installed in accordance with the current regulations on building and safety of use and accessibility [22,23]. Likewise, and as it has movable mechanical elements and electrical systems, this design must comply with current European regulations or those of the country in which it is to be marketed. Thus, standards such as those for the marketing and commissioning of machines, safety in automatic doors, electromagnetic compatibility, and regulations on low voltage installations are applicable [24–27].

Being aware that technology is important in a climate crisis or emergency [28] and that information and communication technologies (ICT) could help address these challenges [29]; for the mechanical design of the door, the lintels, and the dam, the electrical and electronic design of the circuits and the source code for the management of the input and output signals in the microcontroller, respectively, different software have been used:

Technology is important in a climate crisis or emergency [30] and information and communication technologies (ICT) can help to meet these challenges [29]. With this in mind, as shown in the project planning (Figure 1), different phases have been developed for the mechanical design of the door, the lintels and the dam, the electrical and electronic design of the circuits, and the source code for the management of the input and output signals in the microcontroller: (i) Phase I: Solidworks®: This program is used to design the enclosure to house and protect the PLC (Programmer Logic Controller), as well as the different devices of the system; (ii) Phase II: Autocad® electrical: Thanks to its extensive library of electrical and electronic elements, it allows the generation of drawings that will be used later in the manufacture and assembly of the electronic boards: (iii) Arduino® IDE: It is used to write and load programs on the Arduino board. The source code for the IDE is released under the GNU General Public License, version 2; (iv) Tia Portal® (Totally Integrated Automation): This automation and control engineering platform of the German brand Siemens®, offering automation solutions in all industrial sectors software is used to program the PLC, the HMI (Human–Machine Interface) screen, and its applications for Smartphone, as well as the configuration of the servo-drivers for the bar and mobile lintels.

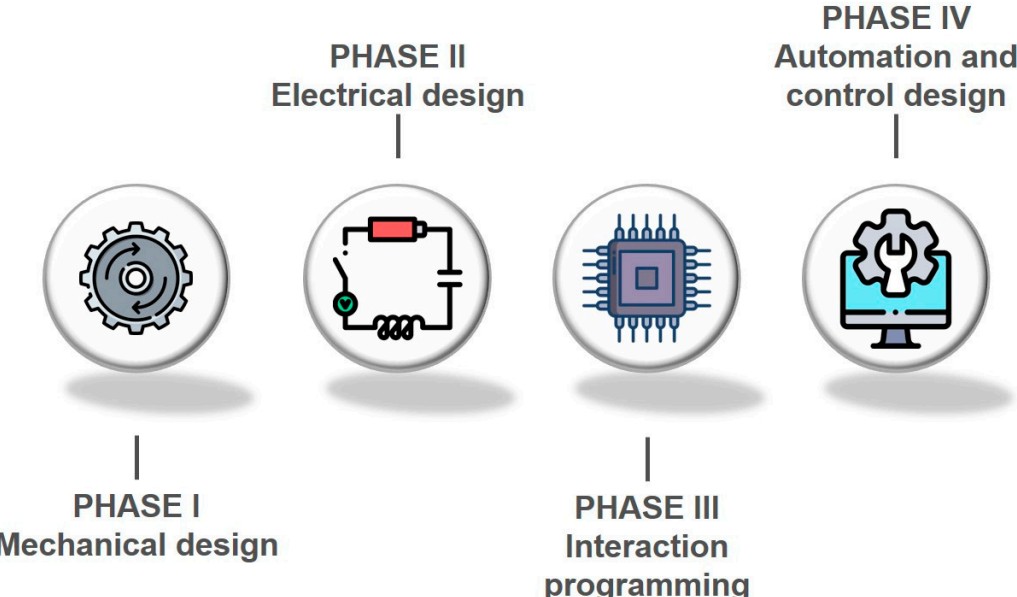

**Figure 1.** Project planning: main phases and ICTs.

### 3. Model Invention

The object of the invention is a device that includes an automatic barrier installed in front of the entrance door to the house. This barrier is responsible for hermetically sealing the door, preventing the passage of water into the house. This barrier is complemented with one or more hinged lintels placed on the side from which the water flow circulates in case of flooding, since this invention is designed for doors placed on streets with a certain slope. This way, these lintels are intended to divert the water towards the center of the street and thereby generate a dry area near the entrance door to the house.

One of the advantages of this barrier is that it does not need to be installed by the user prior to flooding, as is the case with most current systems, because it is a fully automatic system. In case of a possible flood, the system acts intelligently diverting and preventing the entry of water into the building, thus protecting the property and not requiring the presence of the user for installation. However, the system will normally have a manual control so that the user can deploy and collect the lintels and the vertical door barrier if necessary.

Another important feature of this device is that it has a lintel on at least the side of the door through which the water flow is expected to come in the event of a torrential flood. This lintel is at least the height of the barrier and is also automatically activated by swinging outward to divert the flow of water away from the door and the protective barrier, toward the street. For the sake of clarity, a table of components of the invention SFB is presented in the Table 2.

According to the features of the invention, such a barrier moves along lateral guides (3), provided with at least one valve connected to a pneumatic compressor, which is activated when the barrier has reached the highest position, filling the chamber with air until it hermetically closes the front of the gate. As a supplementary but also essential device for the invention (Figure 2), it has been provided with at least one lintel (4) which is installed on the right or left side of the gate depending on the slope of the street or sidewalk (5) in front of the gate (1), since what is intended with such a lintel (4) is to divert the flow of water to the center of the same, diverting it from the door (1) and the barrier (2) to keep a dry area in front of the door so that the water does not fall directly on the barrier (2).

**Table 2.** SFB components.

| Identification Number | Component | Description |
|---|---|---|
| 1 | Door | It is not directly part of the invention, but is fundamental to prevent water from entering the home. |
| 2 | Barrier | Hidden below the ground and which is automatically raised when there is a certain level of water. |
| 3 | Lateral Guides | Fixed to the pillars of the gate to be protected and has a peripheral chamber |
| 4 | Lintel | Installed on the right or left side of the gate depending on the slope of the street or sidewalk |
| 5 | Access | To be protected and which once fully raised closes in a watertight manner totally preventing the passage of water up to the gate |
| 6 | Valves | Connects to a small pneumatic compressor |
| 7 | Chamber | Installed perimetrically to ensure the sealing. |

Note: The gate is not part of the invention.

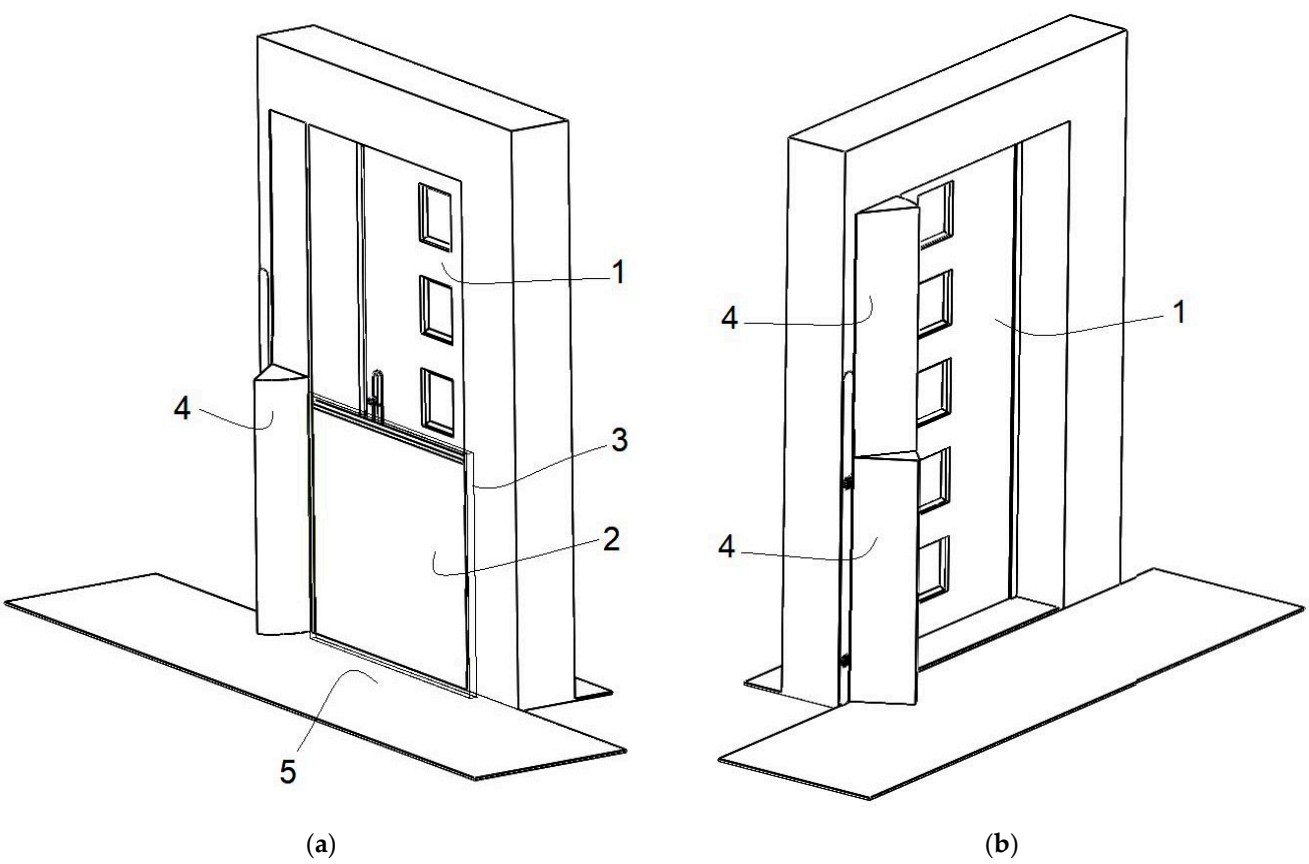

(**a**)  (**b**)

**Figure 2.** View from a point on the street: (**a**) gate (1) with the barrier (2) raised and one of the lintels (4) deployed; (**b**) two overlapping lintels (4) deployed and the barrier (2) hidden, inoperative.

As can be seen in Figure 3, the barrier (2) is mounted on two side-guides (3) and is driven by a servomotor that is activated when at least one Karman sensor (Figure 4)—which is located at the bottom of the door (1) to be protected—is covered by the water and consequently varies its resistance. This indicates, to the PLC, the need to start the ascent of the containment barrier (2) located in front of the door (1), until it reaches the

predetermined height so that the lower part of the same is protected and watertight at street level (5). Karman sensors (Figure 4) detect a liquid flow rate in tubes and pipes, of digital type (all/nothing); they are only activated if it detects fluid circulation by Von Karman effect, valid only for Newtonian fluids. This type of sensor is practically maintenance-free, highly accurate, and has a wide operating range, and its resistance to ingress of solid particles and water is IP67.

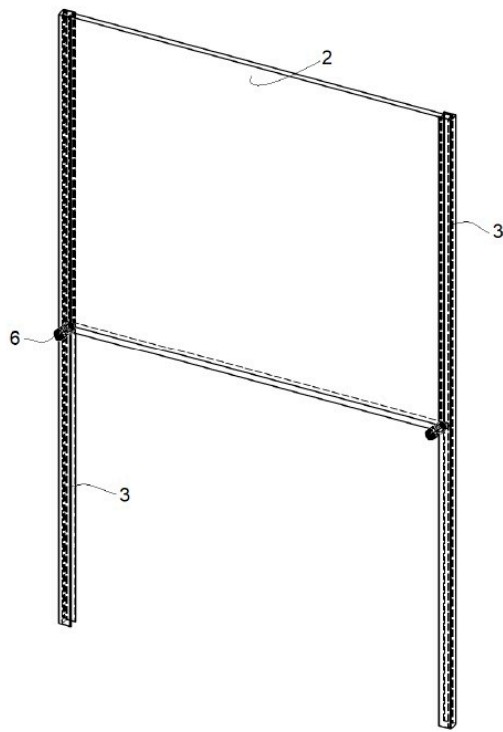

**Figure 3.** View of the barrier (2) mounted on its side-guides (3).

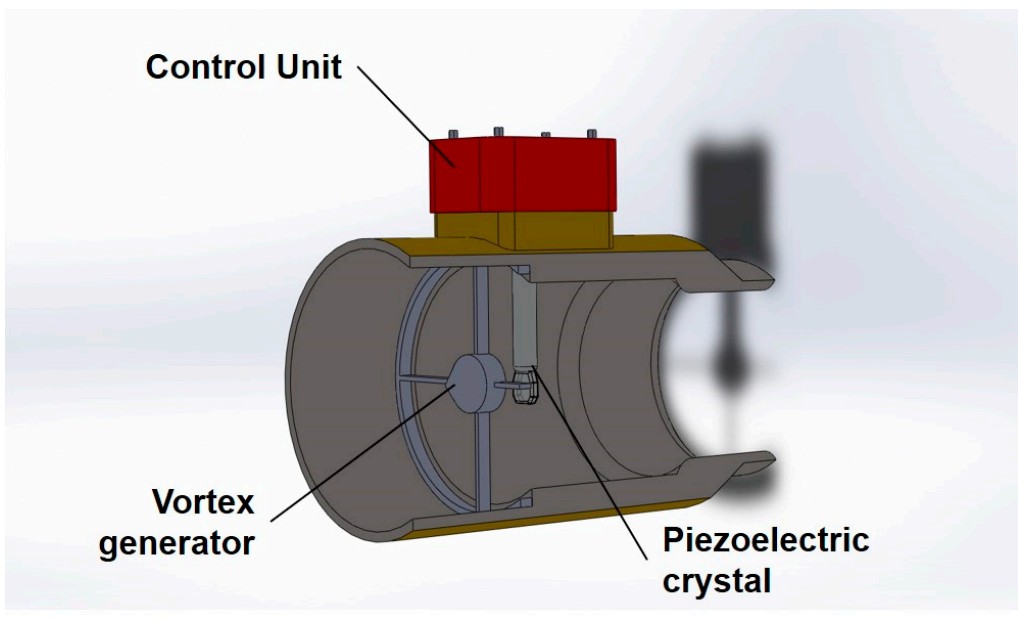

**Figure 4.** Overview of Karman high-precision pressure sensors.

Sensor 1 is placed at 10 cm from the ground level and sensor 2 is placed at 80 cm from the ground level; in the event of an increase in the water level it will circulate through

each tube. In the central part of the pipe, the sensor has a vortex generator and this element causes the fluid flowing through it to become disordered, changing from laminar to turbulent, thus creating differential pressure vortices at the outlet of the pipe. The existence of vortices causes the piezoelectric crystal located next to the generator to oscillate at a frequency producing a voltage, this will be sufficient for the activation of the control system included within the sensor itself, thus sending the digital signal 1 to the PLC input module, causing the activation of the corresponding outputs for barrier and lintels, resulting in their deployment.

In the section illustrated in Figure 5, the barrier (2) presents peripherally a chamber (7) with at least one valve (6) that connects to a small pneumatic compressor, which is, in turn, activated when a detector located in the upper part of the barrier (2), or by means of an incremental encoder. This device will be in charge of sending information to the PLC, indicating that the barrier is in its highest position. To avoid rupture of the chamber (7) and to maintain the air inside the chamber, the pneumatic compressor has a pressure switch that stops the air supply when it reaches a maximum pressure inside the chamber (7). The valve (6) has been foreseen to be a two-way valve, so that it keeps the air inside the chamber (7) until the moment when the barrier (2) is to be collected, then changing its position, and releasing the air contained inside the chamber (7) to the atmosphere.

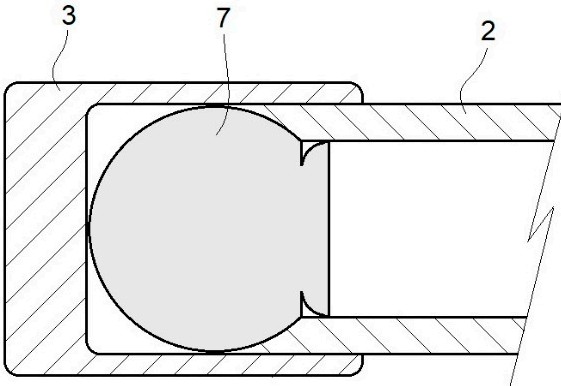

**Figure 5.** Section of the barrier (2) included in its guide (3) and with the chamber (7) inflated to make a hermetic seal in this area.

The airtightness in closing the barrier (2) is achieved by inflating the chamber (7) against the sides of the guides (3), as well as the lower area in which it is hidden (pit) when the damper is retracted, thus preventing the passage of water towards the door of the house and towards the pit itself.

From Figure 6, it can be noticed that the lintel (4) generally has a height equal to or greater than the barrier (2). In some cases, a second lintel could be placed above the previous one, in case the water level rises (in the case of a flood), when it would extend out of the street in the same way as the previous one, helping to divert the water flow towards the center of the street.

As the barriers require real-time and efficient monitoring [1,27], the system is controlled by a PLC that manages the signals coming from the Karman sensors connected to the digital input terminals and located at ground level. In this way, the Karman sensors are located just a few millimeters above the ground, thus determining when the front barrier (2) should start to rise and when the lower hinged lintel (4) should start to be lowered. For situations of larger floods, another Karman sensor is placed approximately at the upper height of the barrier (2), in charge of detecting when it is raised and of activating the upper hinged lintel, when it exists.

Both the hinged lintels (4) and the front barrier (2) are heavy mechanical elements. When they are moving, for the computation of the number of devices to be automatically moved by the invention, the forces produced by the thrust of the fluid to be diverted and contained must be taken into account, in addition to their own weight. To be able to control

the actuating servomotors of these elements, which will be of high power and torque, a relay module is installed. The servomotors are powered by an AC power circuit with high voltage and intensity, in charge of supplying the necessary power to the actuators (barrier and lintel servomotors).

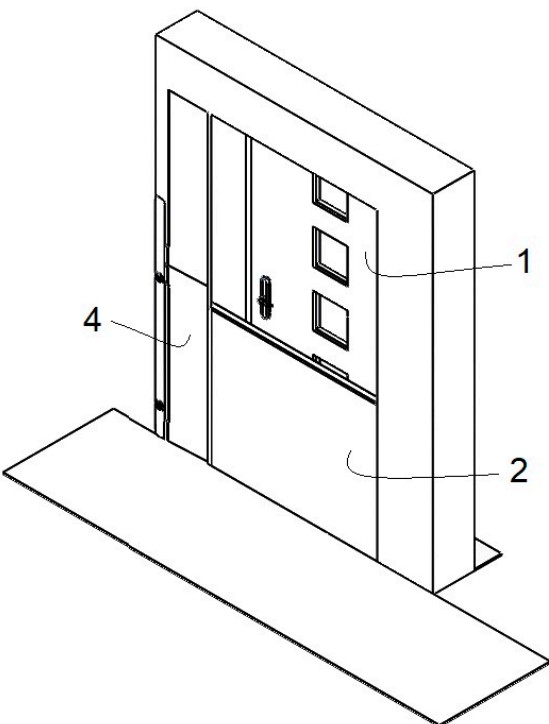

**Figure 6.** From an opposite point of view, the lintels (4) are hidden and yet the barrier (2) is raised.

This relay module is installed in the PLC output module and will be activated by low-voltage and low-current DC (direct current) digital signals. It has been foreseen that each lintel and the door barrier will be moved by servomotors equipped with an incremental encoder, which will be in charge of sending information to the PLC analog inputs of the number of turns of each of the servomotors for the actuation of lintels and front barrier, to stop these actuators once they have completed their travel. In case of loss of power, the supply is based on an Uninterruptible Power Supply (UPS). Through this system, in the event of loss of the mains power supply, the UPS backup batteries are responsible for sending sufficient voltage and current to activate the barrier and the lintels.

The control of the system, as discussed above, is fully automatic, but the user can take control from a console installed in the panel of his home, or remotely, as this console connects via wifi and, therefore, it is possible to take manual control of the system in real time by means of an application installed on a Smartphone. In the same way, the control can be performed from the interior of the house, through an HMI system, being able to force the digital variables of the sensors and thus activating digital output signals to the actuators (lintel and barrier servomotors, in addition to the inflation of the air chamber of the front barrier).

The costs (Table 3) show that a ful SFB costs approximately USD 8000, where the component cost amounts to approximately USD 5150, the assembly cost around USD 1850, and the variable costs are around USD 1000.

**Table 3.** SFB costs.

| Identification Number | Component | Cost (USD) |
|:---:|:---:|:---:|
| 2 | Barrier | 1200 |
| 3 | Lateral Guides | 500 |
| 4 | Lintel | 2000 |
| 5 | Access | 500 |
| 6 | Valves | 250 |
| 7 | Chamber | 100 |
| - | Other components | 1500 |
| - | Variable costs | 1000 |
| - | Assembly cost | 1850 |
| - | Total implementation cost | 8000 |

Elements such as barriers, lintels, and guides will be made of aluminum. The electromechanical components will include servomotors, HMI screen and Siemens PLC S71200. The inflatable chamber around the perimeter of the entrance barrier (7) will be made of butyl, a type of resistant rubber that is easy to expand and can withstand compressed air. The two-way valves (6) and the compressed air supply system for inflation will be a mini-compressor, both from SMC Pneumatics. The rest of the cost is associated with the assembly and adjustment of the system and the variable costs reserved for all problems, such as cost overruns that may arise during the design, assembly, and commissioning. Finally, Figure 7 shows the SFB final 3D design and provides further details of the invention.

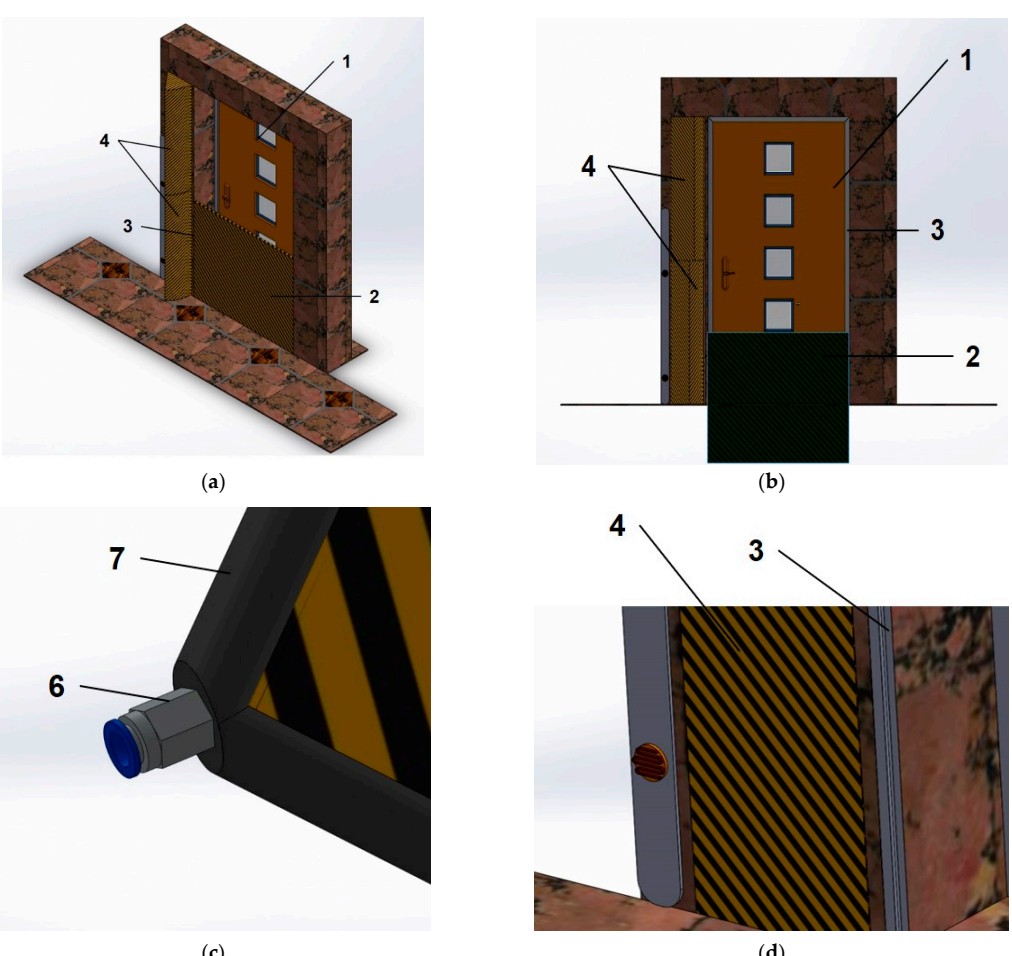

**Figure 7.** Overview final design of Smart Flood Barrier (SFB): (**a**) SFB final 3D design; (**b**) barrier (2) in the upward movement; (**c**) pneumatic valve (6) and chamber (7); (**d**) Karman sensor position.

## 4. Conclusions

Although there are different devices to prevent floodwater from entering the homes, these systems do not work autonomously; they require manual installation by the user. Thus, these systems have the following disadvantages: (i) in the event of an imminent and sudden flood, if the user is not at home, his/her dwelling will be unprotected; and (ii) poor installation by the user may cause water to enter the dwelling.

On the other hand, Smart Flood Barrier (SFB) is an intelligent and automatic system, integrated in the door of the house (in the lintels and in the door's entrance footprint). Thus, the system can work in two modes: (i) autonomously without the supervision of the owner through a system of Karman sensors installed on the lintel or close to it, and (ii) through an application installed on the user's Smartphone, which will allow him to activate the system remotely (even from another location), since it will have a Bluetooth and wifi system, or through an HMI (Human–Machine Interface) screen located inside the house. These modes of operation of the SFB are the differentiating element of this flood barrier design with respect to other models already on the market.

## 5. Patents

This utility mechanism for Smart Flood Barrier has been registered with the following code corresponding to the invention's country of origin, Spain: ES1261399.

**Author Contributions:** Conceptualization, J.M.-C. and D.V.; methodology, J.M.-C.; validation, J.M.-C. and D.V.; investigation, J.M.-C., D.V., P.F.-A. and Á.A.-S.; resources, J.M.-C.; writing—original draft preparation, J.M.-C., D.V., P.F.-A. and Á.A.-S.; writing—review and editing, J.M.-C., D.V., P.F.-A. and Á.A.-S.; supervision, D.V., P.F.-A. and Á.A.-S. All authors have read and agreed to the published version of the manuscript.

**Funding:** This research received the TCUE 2018–2020 award from the Junta de Castilla y León (Spain).

**Institutional Review Board Statement:** Not applicable.

**Informed Consent Statement:** Not applicable.

**Data Availability Statement:** Not applicable.

**Conflicts of Interest:** The authors declare no conflict of interest.

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
