# Peer review of "Design of a Smart Barrier to Internal Flooding"

_inventions, doi:10.3390/inventions7040088_

Round 1

Reviewer 1 Report

General Comments:

The research article is designed to study the “Smart Flood Barrier” inventions with advanced and automated functions.

The authors try to overcome the shortcoming of the previous inventions and try to provide the best solution for an emergency response to the floods to prevent the house.

The discrepancies of the article are mentioned below to improve its quality.

Revisions:

Try to add 1-2 more keywords.

The introduction section is very short so try to increase this section. Moreover, move the figure-1 to the end other picture to show the final design of the door and also label the final picture key parts with the numbering for easy understanding.

While discussing the other parts of the figure, I would recommend adding colors in each part to differentiate it from other parts and using the same color for the same part till the final figure which I have suggested you put the figure-1 at the end as a final figure.

Moreover, explain all the key parts from 1-7 at one place and write their working flow.

The methodology section and invention section needs revisions to make the write-up in line and to make a flow of the article.

Mostly when there is a flood in the area then at that time the electricity is also shut off to avoid great loss, damage or electricity is shut off, or an electric pole is damaged due to the high speed of air then how this system will be activated? Do you have a plan to add the battery backup to start this system? If you did not have a plan then I would recommend adding battery backup.

I would also recommend adding the cost of each part in the table and also quoting the total cost of this automated flood control door.

Have you already made the smartphone application to control this system through a smartphone or it is just your recommendation? If you have already developed then try to add the working principle or communication between two subjects.

Author Response

Please, find enclosed a documente with a detailed response.

Reviewer 2 Report

The present manuscript presents an invention called “Smart Flood Barrier” (SFB). The writing is barely OK with some questionable usage of words (mentioned below) and the reviewer has the following comments:

1) The actual introduction comprises only lines 1-57. After line 57, the authors start to talk about the SFB of the present invention. The introduction does mention several patented flood barriers. However, the introduction fails to discuss the shortcomings of existing techniques. As an academic paper, the introduction should address the void of literature thus the present study/invention is required.

2) Lines 80-113: The reviewer does not think this section is required. How arduous or difficult to make this study/invention work is irrelevant in an academic article.

3) Line 134: The reviewer does not consider “the gate” as part of the invention.

4) Line 138: What does “in C” mean?

5) Line 151: brings some intensity??

6) The device is activated by a light-dependent resistor. Will it be activated by nightfall or even objects at the gate? The current design is not feasible.

7) Line 157-158: Do you want to use a conditional sentence (if …) here?

8) As mentioned in the points above, the writing of the manuscript requires significant improvements. Please have a native speaker review the whole article.

9) Figure 5: Please note that all numbers are upside down.

10) Testing of a prototype should be provided.

11) Automatic flood gates are not new. Many manufacturers, such as https://floodcontrolinternational.com/self-closing-flood-barriers/ , have been making similar products. The current invention is not novel.

Author Response

(The authors gave the same response as above.)

Round 2

Reviewer 1 Report

The authors improved the article as per reviewer comments.

Author Response

The authors are very grateful to the reviewer for his/her assessment of the work and are pleased that he/she liked it.

Reviewer 2 Report

Thanks for providing an improved manuscript. The reviewer appreciates the efforts, and wants to provide a few more comments:

1) A scientific paper is not a patent. A patent can encompass possible variation with simple descriptions, like what the authors did on lines 266-278. Nevertheless, a scientific paper must describe mechanisms in detail. Currently, lines 266-278 provide little detail on how GSM or pressure sensors can be implemented in the design. There is just not enough information.

2) The present invention is not quite different from the automatic flood gate that the reviewer cited in the previous round of review. Both require some mechanical devices to move the floodgates. The only difference is that the present invention utilizes more electronics, but the reviewer does not consider it an advantage. Electronics are more delicate and require more maintenance. The cited device uses simple physics and should be more reliable. The authors must provide a more compelling argument on why the present invention is superior.

Author Response

Thanks for providing an improved manuscript. The reviewer appreciates the efforts, and wants to provide a few more comments:

Round 3

Reviewer 2 Report

The reviewer appreciates the authors for providing a new manuscript and a set of response. The reviewer accepts most of the reasoning. The only thing that requires revising is the "future improvement" part (lines 255-262) of the manuscript. The authors should provide detailed design for the improvement, or delete this part for good. Again, this is not a patent application.

Author Response

Dear reviewer:
The authors are very grateful for your kind appraisal of our work and your instructive recommendation.

Round 4

Reviewer 2 Report

Thanks for providing a revised manuscript. The reviewer believes that the manuscript currently has satisfied the requirement of this journal and can be published.